# WHY DO MULTI-AGENT LLM SYSTEMS FAIL?

**Mert Cemri**[1*], **Melissa Z. Pan**[1*], **Shuyi Yang**[2*], **Lakshya A. Agrawal**[1], **Bhavya Chopra**[1], **Rishabh Tiwari**[1],
**Kurt Keutzer**[1], **Aditya Parameswaran**[1], **Dan Klein**[1], **Kannan Ramchandran**[1],
**Matei Zaharia**[1], **Joseph E. Gonzalez**[1], **Ion Stoica**[1]
[1]University of California, Berkeley
[2]Intenso Sao Paolo
[*]Equal contribution

## ABSTRACT

Despite growing enthusiasm for Multi-Agent LLM Systems (MAS), their performance gains across popular benchmarks often remain minimal compared to single-agent frameworks. This gap highlights the need to systematically analyze the challenges hindering MAS effectiveness.

We present **MAST** (**M**ulti-**A**gent **S**ystem Failure **T**axonomy), the first empirically grounded taxonomy designed to understand MAS failures. We analyze five popular MAS frameworks across over 150 tasks, involving six expert human annotators. Through this process, we identify 14 unique failure modes, organized into 3 overarching categories: (i) specification issues, (ii) inter-agent misalignment, and (iii) task verification. **MAST** emerges iteratively from rigorous inter-annotator agreement studies, achieving a Cohen's Kappa score of 0.88. To support scalable evaluation, we develop a validated LLM-as-a-Judge pipeline integrated with **MAST**. We leverage two case studies to demonstrate **MAST**'s practical utility in analyzing failures and guiding MAS development. Our findings reveal that identified failures require more complex solutions, highlighting a clear roadmap for future research. We open-source our comprehensive dataset and LLM annotator to facilitate further development of MAS[1].

*"Happy families are all alike; each unhappy family is unhappy in its own way."* Tolstoy (1878)

*"Successful systems all work alike; each failing system has its own problems."* (Berkeley, 2025)

## 1 INTRODUCTION

Recently, Large Language Model (LLM) based agentic systems have gained significant attention in the AI community Patil et al. (2023); Packer et al. (2024); Wang et al. (2024a). This growing interest comes from the ability of agentic systems to handle complex, multi-step tasks while dynamically interacting with diverse environments, making LLM-based agentic systems well-suited for real-world problems Li et al. (2023). Building on this characteristic, multi-agent systems are increasingly explored in various domains, such as software engineering Qian et al. (2023); Wang et al. (2024d), drug discoveries Gottweis et al. (2025); Swanson et al. (2024), scientific simulations Park et al. (2023b), and recently general-purpose agent Liang et al. (2025).

Although the formal definition of agents remains a topic of debate Cheng et al. (2024); Xi et al. (2023); Guo et al. (2024a); Li et al. (2024b); Wang et al. (2024b), in this study, we define a LLM-based **agent** as an artificial entity with prompt specifications (initial state), conversation trace (state), and ability to interact with the environments such as tool usage (action). A **multi-agent system** (**MAS**) is then defined as a collection of agents designed to interact through orchestration, enabling collective intelligence. MASs are structured to coordinate efforts, enabling task decomposition, performance parallelization, context isolation, specialized model ensembling, and diverse reasoning discussions He et al. (2024b); Mandi et al. (2023); Zhang et al. (2024); Du et al. (2023); Park et al. (2023a); Guo et al. (2024a).

---

[1]https://github.com/multi-agent-systems-failure-taxonomy/MAST

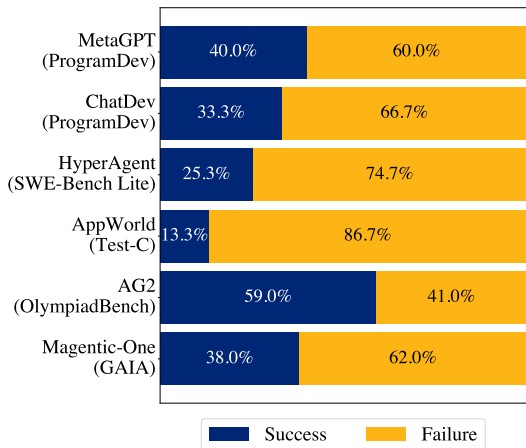

Figure 1: Failure rates of five popular Multi-Agent LLM Systems with GPT-4o and Claude-3.

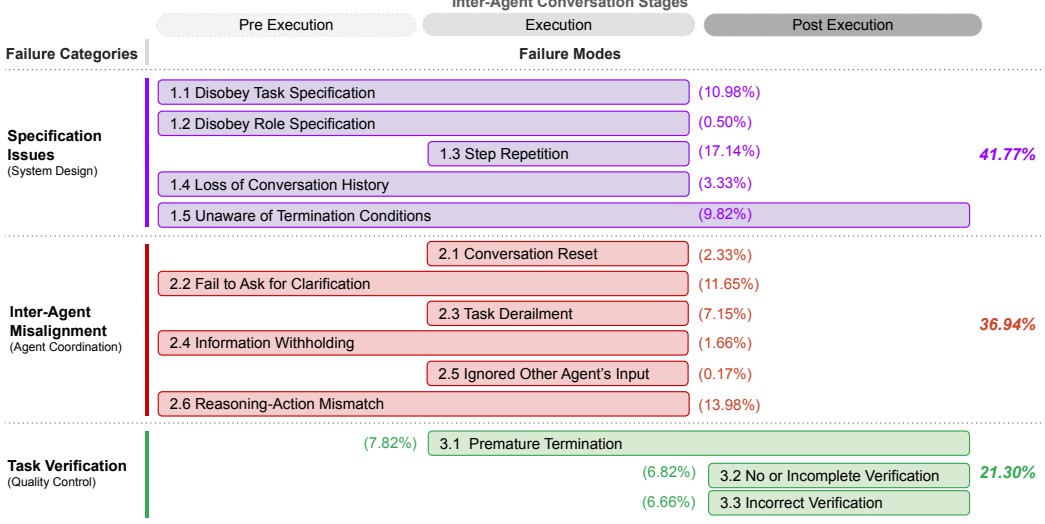

Figure 2: A **Taxonomy of MAS Failure Modes**. The inter-agent conversation stages indicate when a failure can occur in the end-to-end MAS system. If a failure mode spans multiple stages, it means the issue involves or can occur at different stages. Detailed definition and example of each failure mode is available in Appendix A.

Despite increasing adoption of MAS, the gain in accuracy or performance remains minimal compared to single agent frameworks Xia et al. (2024) or even simple baselines such as best-of-N sampling on popular benchmarks Kapoor et al. (2024). Our empirical analysis reveals that the correctness of the state-of-the-art (SOTA) open-source MAS, ChatDev Qian et al. (2023), can be as low as 25%, as shown in Fig. 1. Furthermore, there is no clear consensus on how to build robust and reliable MASs. This leads to a fundamental question that we need to answer first: Why do MASs fail?

To understand MAS failure modes, we conduct the first systematic evaluation of MAS execution traces using Grounded Theory (Glaser & Strauss, 1967). We analyze five popular open-source MASs, employing six expert annotators to identify fine-grained issues across 150 conversation traces, each averaging over 15,000 lines of text. We define failures as cases where the MAS does not achieve the intended task objectives. To ensure consistency in failure modes and definitions, three expert annotators independently label 15 traces, achieving an inter-annotator agreement with a Cohen's Kappa score of 0.88. From this comprehensive analysis, we identify 14 distinct failure modes, which we cluster into 3 primary failure categories. We introduce the **M**ulti-**A**gent **S**ystem

Failure **T**axonomy (**MAST**), the first structured failure taxonomy of MAS, as illustrated in Fig. 2. We do not claim **MAST** covers every potential failure pattern; rather, it serves as the first step towards taxonomizing and understanding MAS failures.

To enable scalable automated evaluation, we introduce an LLM-as-a-judge pipeline Zheng et al. (2023) using OpenAI's o1. We validate this pipeline against expert annotations, achieving a Cohen's Kappa agreement score of 0.77.

To demonstrate **MAST**'s practical usage in guiding MAS development via failure analysis, we conduct case studies involving interventions on improved role specification and architectural changes. We use our LLM annotator to obtain detailed failure breakdowns before and after these interventions, showcasing how **MAST** provides actionable insights for debugging and development. While interventions yield some improvements (e.g., +15.6% for ChatDev), the results show that simple fixes are still insufficient for achieving reliable MAS performance. Mitigating identified failures will require more fundamental changes in system design.

These findings suggest that **MAST** is not merely an artifact of existing multi-agent frameworks, but rather indicative of fundamental design flaws in MAS. Towards building robust and reliable MAS, **MAST** serve as a framework for guiding future research, outlining potential solutions for each of the 14 failure modes. We also open source our annotations for further research on MAS.

While one could simply attribute these failures to limitations of present-day LLM (e.g., hallucinations, misalignment), we conjecture that improvements in the base model capabilities will be insufficient to address the full **MAST**. Instead, we argue that good MAS design requires organizational understanding – even organizations of sophisticated individuals can fail catastrophically (Perrow, 1984) if the organization structure is flawed.

## 2 RELATED WORK

### 2.1 CHALLENGES IN AGENTIC SYSTEMS

The promising capabilities of agentic system has inspired research into specific agentic challenges. For instance, Agent Workflow Memory Wang et al. (2024e) addresses long-horizon web navigation by introducing workflow memory to enhance agent adaptability and efficiency. DSPy Khattab et al. (2023) and Agora Wang et al. (2024e) tackle issues in communication flow, and StateFlow Wu et al. (2024b) focuses on state control within agentic workflows to improve task-solving capabilities. While these works meaningfully contribute towards particular use cases, they do not provide a comprehensive understanding of why MASs fail or propose a strategy that can be broadly applied across domains.

Numerous benchmarks have been proposed to evaluate agentic systems Jimenez et al. (2024); Peng et al. (2024); Wang et al. (2024c); Anne et al. (2024); Bettini et al. (2024); Long et al. (2024). These evaluations are crucial in identifying challenges and limitations in agentic systems, yet they primarily facilitate a top-down perspective, focusing on higher-level objectives such as task performance, trustworthiness, security, and privacy Liu et al. (2023b); Yao et al. (2024b).

### 2.2 DESIGN PRINCIPLE FOR AGENTIC SYSTEMS

Several works highlight the challenges of building robust agentic systems and suggest new strategies, typically for single-agent designs, to improve reliability. For instance, Anthropic's blog Anthropic (2024a) draws the importance of simplicity and modular components, such as prompt chaining and routing, rather than adopting overly complex frameworks. Similarly, Kapoor et al. (2024) shows that complexity can hinder real-world adoption for agentic systems. Our work extends these insights by systematically investigating the failure modes in MASs, offering a taxonomy that demonstrates why MASs fail, and suggesting solutions that align with these insights for agentic system design.

### 2.3 FAILURES TAXONOMIZATION IN LLM SYSTEMS

Despite the growing interest in LLM agents, dedicated research on their failure modes is surprisingly limited. In parallel to Bansal et al. (2024)' study that identifies and organizes challenges in human-

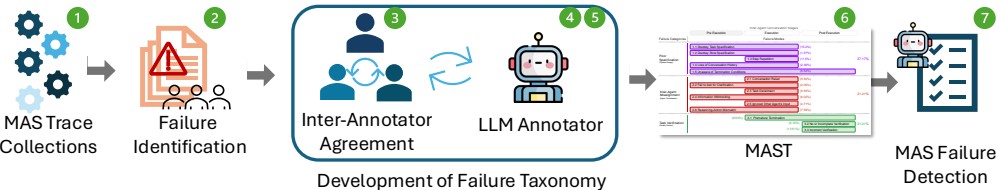

Figure 3: Methodological workflow for systematically studying MAS, involving the identification of failure modes, taxonomy development, and iterative refinement through inter-annotator agreement studies by achieving a Cohen's Kappa score of 0.88.

Table 1: Table of MASs studied with at least 30 human-annotated traces. Details and other systems can be found in Appendix B.

| Multi-Agent System | Agentic Architecture | Purpose of the System |
|---|---|---|
| MetaGPT Hong et al. (2023) | Assembly Line | Simulating the SOPs of different roles in Software Companies to create open-ended software applications |
| ChatDev Qian et al. (2023) | Hierarchical Workflow | Simulating different Software Engineering phases like (design, code, QA) through simulated roles in a software engineering company |
| HyperAgent Phan et al. (2024) | Hierarchical Workflow | Simulating a software engineering team with a central Planner agent coordinating with specialized child agents (Navigator, Editor, and Executor) |
| AppWorld Trivedi et al. (2024) | Star Topology | Tool-calling agents specialized to utility services (ex: GMail, Spotify, etc.) being orchestrated by a supervisor to achieve cross-service tasks |
| AG2 Wu et al. (2024a) | N/A - Agentic Framework | An open-source programming framework for building agents and managing their interactions. |

agent interaction in agentic system, our contribution represents a pioneering effort in studying failure modes in MASs. This highlights the need for future research in developing robust evaluation metrics, identifying common failure patterns, and designing mitigation strategies to improve the reliability of LLM agents.

## 3 STUDY METHODOLOGY

This section describes our methodology for identifying dominant failure patterns in MAS and establishing a structured taxonomy of failure modes. Figure 3 provides an overview of this workflow.

First, we would like to note that gathering and proposing a taxonomy of failure modes is a highly nontrivial task that requires significant effort and consideration: the taxonomy should be broad enough to cover different kinds of failure modes that may arise in diverse MASs and benchmarks, but also specific and detailed enough to offer insights into the failures observed. Moreover, when multiple people use the taxonomy to classify the failures in a MAS execution, the different conclusions should largely agree, which means that the taxonomy should yield a crystal clear understanding of what different failure modes mean.

To systematically uncover failure patterns without bias, we adopt the **Grounded Theory** (GT) approach (Glaser & Strauss, 1967), a qualitative research method that constructs theories directly from empirical data rather than testing predefined hypotheses. The inductive nature of GT allows the identification of the failure mode to emerge organically. We collect and analyze MAS execution traces iteratively with *theoretical sampling*, *open coding*, *constant comparative analysis*, *memoing*, and *theorizing*, detailed in Section 3.1. In total, the GT analysis accross 150+ traces require over 20 hours of pure annotation per annotator who has experience with agentic systems.

After obtaining the MAS traces and discussing our initial findings, we derive a preliminary taxonomy by gathering observed failure modes. To refine the taxonomy, we conduct inter-annotator agreement studies, iteratively adjusting the failure modes and the failure categories by adding, removing, merging, splitting, or modifying the definition until consensus is reached. This process mirrors a *learning* approach, where taxonomy refinement continues until achieving stability, measured by inter-annotator agreement (IAA) through Cohen's Kappa score. To that end, we conduct three rounds of IAA experiments, that require about 10 hours in total, which is solely for resolving the disagreements between annotations, not counting the annotation time itself.

In addition, to enable automated failure identification, we develop an LLM-based annotator and validate its reliability.

## 3.1 DATA COLLECTION AND ANALYSIS

We employ **theoretical sampling** (Draucker et al., 2007) to ensure diversity in the identified MASs, and the set of tasks on which to collect data (MAS execution traces). This approach guided the selection of MASs based on variations in their objectives, organizational structures, implementation methodologies, and underlying agent personas. For each MAS, tasks were chosen to represent the system's intended capabilities rather than artificially challenging scenarios. For instance, if a system reported performance on specific benchmarks or datasets, we selected tasks directly from these benchmarks. The MASs analyzed here span multiple domains and are the first five MAS frameworks in Table 1 and Appendix B. This diversity guided by theoretical sampling ensured a comprehensive exploration of MAS behaviors across contexts.

Upon collecting the MAS traces, we apply **open coding** Khandkar (2009) to analyze the traces we collected for agent–agent and agent–environment interactions. Open coding breaks qualitative data into labeled segments, allowing annotators to create new codes and document observations through memos, which enable iterative reflection and collaboration among annotators. During open coding, we engage in **constant comparative analysis**, systematically comparing new codes with existing ones to refine our understanding. This iterative process continues until we reached **theoretical saturation**, the point at which no new insights emerged from additional data. Through this process, the annotators annotated 150+ traces spanning 5 MASs. Next, we apply **axial coding** Vollstedt & Rezat (2019), grouping related open codes to reveal the fine-grained failure modes in **MAST**. Finally, we identify patterns and link failure modes, ultimately forming a taxonomy of error categories shown in Figure 2.

## 3.2 INTERANNOTATOR AGREEMENT STUDY AND ITERATIVE REFINEMENT

Inter-annotator studies mainly target validating a given test or rubric, such that when multiple different annotators annotate the same set of test cases based on the same rubric, they should arrive at the same conclusions. Even though we initially derive a taxonomy as a result of our theoretical sampling and open coding as explained in the previous section, there still exists the need to validate the non-ambiguity of this taxonomy.

For the inter-annotator agreement (IAA) study, we conduct three major rounds of discussions on top of the initial derivation of taxonomy. In Round 1, we sample 5 different MAS traces from over 150 traces we obtained with theoretical sampling as explained in the previous section, and the three annotators annotate these traces using the failure modes and definitions in the initial taxonomy. We observe that the agreement reached at Round 1 is very weak between annotators, with a Cohen's Kappa score of 0.24. Next, these annotators work on the taxonomy to refine it. This involves iteratively changing the taxonomy until we converge to a consensus regarding whether each and every failure mode existed in a certain failure mode or not in all 5 of the collected traces. In iterative refinement, we change the definitions of failure modes, break them down into multiple fine grained failure modes, merge different failure modes into a new failure mode, add new failure modes or erase the failure modes from the taxonomy, as needed.

This process can be likened to a *learning* study where different agents (this time human annotators) independently collect observations from a shared state space and share their findings with each other to reach a consensus Lalitha et al. (2018). Moreover, in order not to fall into the fallacy of using training data as test data, when we do the refinement studies at the end of Round 1, we test the new

inter-annotator agreement and the performance of the taxonomy in a different set of traces, in Round 2. In the next stage (Round 2), we sample another set of 5 traces, each from a different MAS. Then, the annotators agred substantially well on the first try, attaining an average Cohen's Kappa score of 0.92 among each other. Motivated by this, we proceed to Round 3, where we sampled another set of 5 traces and again annotated using the same finalized taxonomy, where achieved an average Cohen's Kappa score of 0.84. Note that Cohen's Kappa score of more than 0.8 is considered strong and more than 0.9 is considered almost perfect alignment McHugh (2012).

Motivated by the reliability of our taxonomy, we ask the following question: can we come up with an automated way to annotate traces such that developers or users can use this automated pipeline with our taxonomy to understand the failure reasons of their models? Thus, we developed an automated **MAST** annotator using an LLM-as-a-judge pipeline, which we describe in Section 3.3.

Table 2: Performance of LLM-as-a-judge pipeline

| Model | Accuracy | Recall | Precision | F1 | Cohen's $\kappa$ |
|---|---|---|---|---|---|
| o1 | 0.89 | 0.62 | 0.68 | 0.64 | 0.58 |
| o1 (few shot) | 0.94 | 0.77 | 0.833 | 0.80 | 0.77 |

## 3.3 LLM ANNOTATOR

After developing our taxonomy, **MAST** and completing the inter-annotator agreement studies, we aim to come up with an automated way to discover and diagnose the failure modes in MAS traces using our taxonomy. To that end, we develop an LLM-as-a-judge pipeline. In this strategy, we provide a system prompt to LLMs where we include the failure modes in our **MAST**, their detailed explanation, as shown in Appendix A, and some examples of these failure modes as shown in Appendix D. In that strategy, we decide to use OpenAI's o1 model, and we experiment with both the cases where we do not provide the aforementioned examples (called o1 in Table 2) and where we provide the examples (called o1 few-shot in Table 2. Based on the results of Round 3 of inter-annotator agreement study mentioned in Section 3.2, we test the success of the LLM annotator, as shown in Table 2. As we achieve an accuracy of 94% and a Cohen's Kappa value of 77%, we deem that the LLM annotator, with in context examples provided, to be a reliable annotator. Motivated by this result, we let the LLM annotator annotate the rest of the traces in the 150+ trace corpora we gathered, the result of which are shown in Figure 4, and the final taxonomy with the distribution of failure modes is shown in Figure 2.

## 4 STUDY FINDINGS

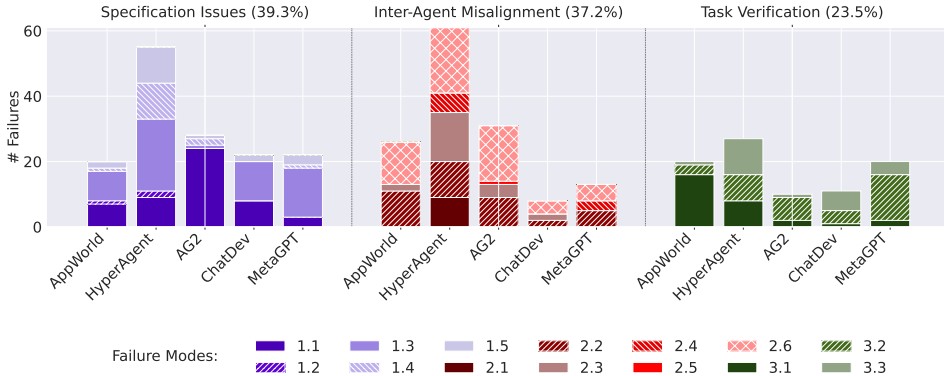

Figure 4: Distribution of failure modes by categories and systems. Since failures are detected on different tasks, the results are not directly comparable across MASs in a quantitative sense. However, for each MAS, we can analyze how failures are distributed across the three main categories and among the 14 specific failure modes.

We present the Multi-Agent System Failure Taxonomy (**MAST**), shown in Figure 2. We develop the taxonomy through empirical analysis of 150 MAS execution traces across 5 task domains, using Grounded Theory and iterative refinement via inter-annotator agreement studies.

**MAST** identifies 14 fine-grained failure modes, mapping them to execution stages (Pre-Execution, Execution, Post-Execution) where their root causes typically emerge. It organizes these modes into 3 overarching categories based on the fundamental nature of failures.

We propose **MAST** as the first foundational framework for unifying MAS failures. We recognize that prior works have observed some individual modes and do not claim exhaustive coverage, rather, **MAST** offers precise definitions, clear boundaries between failure patterns, and serves a structured approach to understanding challenges in MAS.

## 4.1 MULTI-AGENT SYSTEM FAILURE TAXONOMY

This section presents the failure categories (FC) in **MAST** and discusses their implications. Appendix A provides detailed definitions for each of the 14 fine-grained failure modes (FM), while Appendix D presents concrete examples for each mode.

> **FC1. Specification Issues.** Failures originate from system design decisions, and poor or ambiguous prompt specifications.

Failures in FC1 often manifest during execution but reflect flaws in pre-execution design choices regarding system architecture, prompt instructions, or state management. Failure modes include fail to follow task requirements (FM-1.1, 10.98%) or agent roles (FM-1.2, 0.5%), step repetitions (FM-1.3, 17.14%) due to rigid turn configurations, context loss (FM-1.4, 3.33%), or failing to recognize task completion (FM-1.5, 9.82%).

Failures to follow specifications (FM-1.1 and FM-1.2) are two commonly observed failure modes in **MAST**. Although it may fall under the broad umbrella of a well-known challenges, instruction following, in LLM-based MAS applications, we believe that there exist deeper underlying causes of failure, with different potential fixes: (1) flaws in MAS designs with agent roles and workflow phases, (2) poor user prompt specifications, (3) limitation of the underlying LLM in understanding the instructions, (4) the LLM understanding the instruction but failing to follow the instruction. We posit that a well-designed MAS should be able to interpret task objective from high-level specification containing reasonably inferable details, reducing the need for long-run user prompt via improvement on MAS as a core goal of agentic systems is agency.

For example, a task for ChatDev is to create a Wordle game with the prompt *a standard wordle game by providing a daily 5-letter...}*. The generated program uses a small, fixed word dictionary, failing to infer the daily changing word requirement implied by "standard" and "daily". To demonstrate this extends beyond user prompt ambiguity, we provide a more explicit prompt: *... without having a fixed word bank, and randomly select a new 5-letter word each day*. Despite this clarification, ChatDev still produces code with a fixed word list and introduces new errors (e.g., accept error inputs). Thus, this suggests failures stem from the MAS's inherent design for interpreting specifications.

Despite challenges for LLM in instruction following, we show promising headroom for improving MAS via better system design. We conduct intervention studies to improve agent role specifications (Appendix F). Our studies yield a notable +9.4% increase in success rate for ChatDev, when running on the same user prompt and base LLM (GPT-4o).

> **FC2. Inter-Agent Misalignment.** Failures arise from breakdowns in inter-agent interaction and coordination during execution.

FC2 covers failures in agent coordination that prevent effective agent-agent alignment towards a common goal. Failure modes include unexpected conversation resets (FM-2.1, 2.33%), proceeding with wrong assumptions instead of seeking clarification (FM-2.2, 11.65%), task derailment (FM-2.3, 7.15%), withholding crucial information (FM-2.4, 1.66%), ignoring inputs from other agents (FM-2.5, 0.17%), or mismatches between reasoning and action (FM-2.6, 13.98%). Figure 5 shows an example of information withholding (FM-2.4), where an agent identifies necessary information

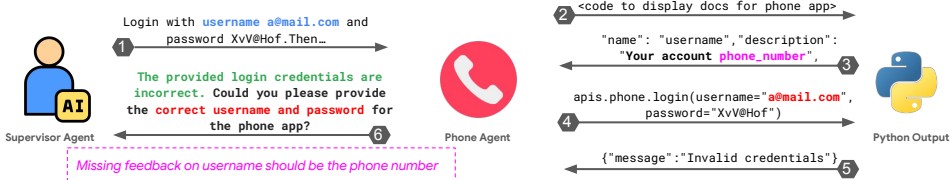

Figure 5: Example of FM-2.4 Information Withholding. The Phone Agent fails to communicate API requirements (username format) to the Supervisor Agent. The Supervisor also fails to seek clarification. Repeated failed login attempts lead to task failure.

(correct username format) but fails to communicate it, leading to repeated failed attempts by another agent, and ultimate failing to complete the task.

Diagnosing FC2 failures can be complex, as different root causes may produce similar surface behaviors. For example, missing information might result from withholding (FM-2.4), ignoring input (FM-2.5), long context length Liu et al. (2023a) or context mismanagement (FM-1.4). Distinguishing these necessitates the fine-grained modes in **MAST**.

> **FC3. Task Verification.** Failures involve inadequate verification processes that fail to detect or correct errors, or premature termination of tasks.

FC3 failures relate to final output quality control. These include premature termination (FM-3.1, 7.82%), no or incomplete verification (FM-3.2, 6.82%), or incorrect verification (FM-3.3, 6.66%). FC3 highlight challenges in ensuring the final output's correctness and reliability. As an example of FM-3.2, a ChatDev-generated chess program passes all rounds of verifications but contains runtime bugs (e.g., accepting invalid moves) because the verifier performs only superficial checks such as code compilation or comments, failing to validate against actual game rules or available online knowledge. This inadequacy persists despite explicit review phases, making the generation output unusable.

## 4.2 MAST Effectiveness Evaluation

We evaluate **MAST**'s effectiveness based on three key aspects: its generalization to unseen systems and datasets, the balanced distribution of identified failures, and the distinctiveness of its failure categories.

**Balanced Distribution.** The distribution of failures across **MAST**'s categories is relatively balanced (FC1: 41.77%, FC2: 36.94%, FC3: 21.30%, Figure 2). The absence of a single dominant category suggests **MAST** provides balanced coverage and captures diverse failure types, rather than reflecting biases from specific system designs. Furthermore, the distinct failure profiles observed across different MAS (Figure 4) highlight **MAST**'s ability to capture system-specific characteristics, such as AppWorld suffers with premature terminations (FM-3.1) and OpenManus suffers from step repetition (FM-1.3).

**Distinct Failure Categories.** Correlation analysis between the main failure categories (Figure 6) shows low correlations (0.17-0.32). This suggests that the categories capture distinct aspects of MAS failures with limited overlap, supporting the taxonomy's structure. This distinctiveness is crucial because, as noted in Insight 2, failures with similar surface behaviors can stem from different root causes (e.g., memory management vs. agent coordination).

Although **MAST**'s fine-grained nature helps differentiate root cause, it also poses a challenge for our LLM annotator. Analyzing correlations between specific failure modes (see Appendix C for Figure 7) shows moderate correlations (max of 0.63) between modes with similar symptoms might lead automated evaluators to conflate distinct root causes.

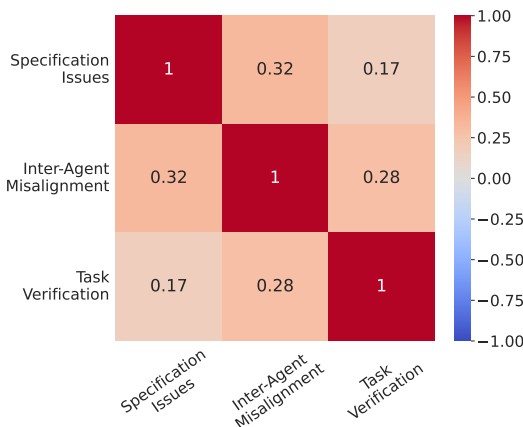

Figure 6: MAS failure categories correlation matrix.

### 4.3 OPEN CHALLENGES BEYOND CORRECTNESS

While developing MAST, we focused primarily on failures related to task correctness and completion, as this is a fundamental prerequisite for usable MAS. However, we observe a significant prevalence of inefficiencies in MAS traces, which MAST currently does not include by design.

Agents often engage in unnecessarily long conversations or take circuitous routes to achieve a goal. For example, in one AppWorld trace, the task was to retrieve the first 10 songs from a playlist. The orchestrator and Spotify agent engaged in 10 rounds of conversation, retrieving one song at a time, even though the Spotify agent's capability allowed retrieving all 10 songs in a single, valid action. Such inefficiencies can lead to dramatically increased costs (token usage) and latency (runtime), sometimes by factors of 10x or more. Addressing this requires optimizing not just for correctness but also for efficiency, cost, and speed.

We deliberately pruned non-correctness metrics like efficiency during **MAST**'s iterative refinement (Section 3 ) to maintain focus. However, we recognize that efficiency, along with other important dimensions like cost, robustness, scalability, and security, are critical for real-world MAS deployment. Developing taxonomies and evaluation methods for these aspects remains important future work.

## 5 TOWARDS BETTER MULTI-AGENT LLM SYSTEMS

Having presented **MAST**, we now discuss its broader implications and utility. **MAST** is not merely a list of definitions; it serves as a foundational framework and practical tool for understanding, debugging, and ultimately improving MAS. By concretely defining failure modes, **MAST** outlines the challenges in building reliable MAS, thereby opening up targeted research problems for the community. This section highlights how **MAST** aids agentic system development, suggesting that progress requires focusing on system design alongside model capabilities.

### 5.1 **MAST** AS A PRACTICAL DEVELOPMENT TOOL

Developing robust MAS presents significant challenges. When a system exhibits a high failure rate on a benchmark (e.g., 75% failure for ChatDev on ProgramDev, Figure 1), pinpointing the underlying causes is difficult, especially if failure manifestations vary widely. Without a systematic framework, developers often resort to ad-hoc debugging of individual failed traces Fritzson et al. (1992). Furthermore, evaluating the impact of interventions is complex; a modest improvement in overall success rate (e.g., +10%) might obscure whether the fix addressed the intended issues, introduced new problems, or only work for specific cases.

Here, **MAST** offers practical value. By providing a structured vocabulary and clear definitions for distinct failure modes, it enables systematic diagnosis. When combined with automated analysis

tools, such as our LLM annotator, developers can obtain a breakdown of failure types occurring in their system across many traces. This quantitative overview pinpoints the most frequent failure modes, guiding debugging efforts towards the highest-impact areas. For example, Fig. 4 suggests that HyperAgent could benefit significantly from addressing its dominant failure modes: step repetition (FM-1.3) and incorrect verification (FM-3.3).

Moreover, **MAST** facilitates rigorous evaluation of improvements. Instead of relying solely on aggregate success rates, developers can perform before-and-after comparisons using **MAST**. Our case studies (Appendix F) illustrate this: applying interventions to ChatDev and AG2 resulted in overall performance gains (Table 4), but a **MAST**-based analysis (detailed in Appendix F.3) reveals which specific failure modes were mitigated and whether any trade-offs occurred (e.g., reducing one failure type while inadvertently increasing another). This detailed view is crucial for understanding *why* an intervention works and for iterating effectively towards more robust systems.

## 5.2 BEYOND MODEL CAPABILITIES: THE PRIMACY OF SYSTEM DESIGN

While one might attribute the observed errors in **MAST** solely to model incapability, a key finding from our intervention studies highlights that many MAS failures came from system design, not just limitations of the underlying LLMs (e.g., hallucination or basic prompt following). Although improved models are beneficial, our results suggest that they are insufficient alone to guarantee reliable MAS performance.

In our intervention case studies (Appendix F), we apply two strategies, architectural (i.e. targeting underlying the topology of the MAS) and prompt modifications inspired by **MAST**'s failure patterns, to improve role adherence and verification, shown in Table 4. To have a fair evaluation, we evaluate MAS with the same LLM and user prompt before and after interventions. The improvement strongly suggests that improvement to the MAS system design itself can reduce failures, independent of base model improvements, underscoring that observed failures are not solely due to model limitations - just like humans can make mistake and have organizational issues with human-level intelligence.

However, these improvements also demonstrate a deeper challenge. While the interventions cause a statistically significant improvement in results, not all failure modes are eradicated, and task completion rates either marginally improved on the tasks that were already good or still remain low indicating that non-trivial improvements are needed. Achieving high reliability likely requires more fundamental changes to agent organization, communication protocols, context management, and verification integration, concepts echoed in studies of complex systems and high-reliability human organizations and more detailed in Table 3. **MAST** provides the necessary framework to identify where these structural weaknesses lie and guide the design and evaluation of more sophisticated MAS architectures. Understanding the root causes pinpointed by **MAST** is essential for designing effective interventions, moving beyond treating symptoms towards addressing core design flaws.

## 6 CONCLUSION

In this study, we present the first systematic investigation of failure modes of LLM based Multi-Agent Systems, where we collect and analyze more than 150+ traces with the guidance of Grounded Theory. We identify 14 fine-grained failure modes, and group them under 3 different failure categories, providing a rubric for future research in MAS. We also provide an automated annotator to diagnose the failure modes in a MAS given our taxonomy, and show the utility of **MAST** as a debugging tool.

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

## ORGANIZATION OF APPENDIX

The appendix is organized as follows: in Section A further details about failure categories and failure modes are given, in Section B we provide some details about the multi-agent systems we have annotated and studied, in Section C we plot the correlations between MAS failure modes, in Section D examples of every failure mode are reported and commented, in Section E we discuss some tactical approaches and structural strategies to make MASs more robust to failures, in Section F we present two case studies where we show that tactical approaches can get only limited results, in Sections G and H there are prompt interventions we tested on AG2 and ChatDev case studies.

## A **MAST** FAILURE CATEGORIES: DEEP DIVE

### A.1 FC1. SPECIFICATION ISSUES

This category includes failures that arise from deficiencies in the design of the system architecture, poor conversation management, unclear task specifications or violation of constraints, and inadequate definition or adherence to the roles and responsibilities of the agents.

We identify five failure modes under this category:

- FM-1.1: **Disobey task specification** - Failure to adhere to the specified constraints or requirements of a given task, leading to suboptimal or incorrect outcomes.

- FM-1.2: **Disobey role specification** - Failure to adhere to the defined responsibilities and constraints of an assigned role, potentially leading to an agent behaving like another.

- FM-1.3: **Step repetition** - Unnecessary reiteration of previously completed steps in a process, potentially causing delays or errors in task completion.

- FM-1.4: **Loss of conversation history** - Unexpected context truncation, disregarding recent interaction history and reverting to an antecedent conversational state.

- FM-1.5: **Unaware of termination conditions** - Lack of recognition or understanding of the criteria that should trigger the termination of the agents' interaction, potentially leading to unnecessary continuation.

## A.2    FC2. INTER-AGENT MISALIGNMENT

This category includes failures arising from ineffective communication, poor collaboration, conflicting behaviors among agents, and gradual derailment from the initial task.

We identify six failure modes under this category:

- FM-2.1: **Conversation reset** - Unexpected or unwarranted restarting of a dialogue, potentially losing context and progress made in the interaction.

- FM-2.2: **Fail to ask for clarification** - Inability to request additional information when faced with unclear or incomplete data, potentially resulting in incorrect actions.

- FM-2.3: **Task derailment** - Deviation from the intended objective or focus of a given task, potentially resulting in irrelevant or unproductive actions.

- FM-2.4: **Information withholding** - Failure to share or communicate important data or insights that an agent possess and could impact decision-making of other agents if shared.

- FM-2.5: **Ignored other agent's input** - Disregarding or failing to adequately consider input or recommendations provided by other agents in the system, potentially leading to suboptimal decisions or missed opportunities for collaboration.

- FM-2.6: **Reasoning-action mismatch** - Discrepancy between the logical reasoning process and the actual actions taken by the agent, potentially resulting in unexpected or undesired behaviors.

## A.3    FC3. TASK VERIFICATION

This category includes failures resulting from premature execution termination, as well as insufficient mechanisms to guarantee the accuracy, completeness, and reliability of interactions, decisions, and outcomes.

We identify three failure modes under this category:

- FM-3.1: **Premature termination** - Ending a dialogue, interaction or task before all necessary information has been exchanged or objectives have been met, potentially resulting in incomplete or incorrect outcomes.

- FM-3.3: **No or incomplete verification** - (partial) omission of proper checking or confirmation of task outcomes or system outputs, potentially allowing errors or inconsistencies to propagate undetected.

- FM-3.3: **Incorrect verification** - Failure to adequately validate or cross-check crucial information or decisions during the iterations, potentially leading to errors or vulnerabilities in the system.

## B  MULTI-AGENT SYSTEMS STUDIED WITH HUMAN-ANNOTATED TRACES

In this section, we provide some more details on MAS we annotated during our study.

### B.1  MAS WITH AT LEAST 30 HUMAN ANNOTATED TRACES

***MetaGPT.*** MetaGPT Hong et al. (2023) is a multi-agent system that simulates a software engineering company and involves agents such as a Coder and a Verifier. The goal is to have agents with domain-expertise (achieved by encoding Standard Operating Procedures of different roles into agents prompts) collaborativelty solve a programming task, specified in natural language.

***ChatDev.*** ChatDev is a generalist multi-agent framework that initializes different agents, each assuming common roles in a software-development company Qian et al. (2024). The framework breaks down the process of software development into 3 phases: design, coding and testing. Each phase is divided into sub-tasks, for example, testing is divided into code review (static) and system testing (dynamic). In every sub-task, two agents collaborate where one of the agents acts as the orchestrator and initiates the interaction and the other acts as an assistant to help the orchestrator achieve the task. The 2 agents then hold a multi-turn conversation to achieve the goal stated by the orchestrator ultimately leading to the completion of the task, marked by a specific sentinel by either agents. ChatDev has the following agent roles: CEO, CTO, Programmer, Reviewer and Tester. ChatDev introduces "Communicative Dehallucination", which encourages the assistant to seek further details about the task over multiple-turns, instead of responding immediately.

***HyperAgent.*** HyperAgent Phan et al. (2024) is a framework for software engineering tasks organized around four primary agents: Planner, Navigator, Code Editor, and Executor. These agents are enhanced by specialized tools, designed to provide LLM-interpretable output. The Planner communicates with child agents via a standardized message format with two fields: Context (background and rationale) and Request (actionable instructions). Tasks are broken down into subtasks and published to specific queues. Child agents, such as Navigator, Editor, and Executor instances, monitor these queues and process tasks asynchronously, enabling parallel execution and significantly improving scalability and efficiency. For example, multiple Navigator instances can explore different parts of a large codebase in parallel, the Editor can apply changes across multiple files simultaneously, and the Executor can run tests concurrently, accelerating validation.

***AppWorld.*** AppWorld is a benchmark, that provides an environment with elaborate mocks of various everyday services like eShopping Website, Music Player, Contacts, Cost-sharing app, e-mail, etc Trivedi et al. (2024). The benchmark consists of tasks that require executing APIs from multiple services to achieve the end-users tasks. The AppWorld benchmark provides a ReAct based agent over GPT-4o as a strong baseline. We create a multi-agent system over AppWorld derived from the baseline ReAct agent, where each agent specializes in using one of the services mocked in AppWorld, with detailed instructions about the APIs available in that service, and access to the documentation for that specific service. A supervisor agent receives the task instruction to be completed, and can hold one-on-one multi-turn conversations with each of the service-specific agents. The service-agents are instructed to seek clarification with the supervisor, whenever required. The supervisor agent holds access to various information about the human-user, for example, credentials to access various services, name, email-id and contact of the user, etc, which the service-agents need to access the services, and must clarify with the supervisor agent.

***AG2.*** AG2 (formerly AutoGen) Wu et al. (2023) is an open-source programming framework for building agents and managing their interactions. With this framework, it is possible to build various flexible conversation patterns, integrating tools usage and customizing the termination strategy.

### B.2  MAS WITH AT LEAST 5 HUMAN ANNOTATED TRACES

***AutoKaggle.*** AutoKaggle is a multi-agent framework designed to solve data science competitions, popularly held on Kaggle, autonomously Li et al. (2024c). Similar to ChatDev above, AutoKaggle has a phase-based workflow. It divides the data science competition process into six key phases: background understanding, preliminary exploratory data analysis, data cleaning (DC), in-depth exploratory data analysis, feature engineering (FE), and model building, validation, and prediction (MBVP). AutoKaggle consists of 5 specialized agents: Reader, Planner, Developer, Reviewer and

Summarizer. In each phase, a subset of these agents are active and work in sequence to complete the phase. The reader agent finds information relevant to the task, by reading the summary from the previous phase and makes observations about the current phase and includes them in an overview. The planner uses the overview to generate a plan to complete the current phase. Next, the developer agent uses tools like code execution, debugger and unit tests to write the code. AutoKaggle also provides a comprehensive set of machine learning tools, abstracting away complex code that would be required to perform compound data processing tasks like "FillMissingValues" into simple API calls that AutoKaggle agents can generate. The reviewer then provides feedback. Finally, the summarizer agent writes a detailed summary of the phase execution including changes (addition/deletions) to the data, and this summary is then passed to the next phase.

***Multi-Agent Peer Review.*** Multi-Agent Peer Review Xu et al. (2023) is a collaboration strategy where each agent independently constructs its own solution, peer-reviews the solutions of others, and assigns confidence levels to its reviews. Upon receiving peer reviews, agents revise their initial solutions, and the final prediction is determined through a majority vote among the n participating agents.

***MA-ToT.*** Multi-Agent Tree of Thoughts leverage the strengths of both multi-agent reasoning and Tree of Thoughts (ToT) strategies. In this system, multiple Reasoner agents operate in parallel, employing ToT to explore diverse reasoning paths. Then, a Thought Validator verifies these paths and promotes valid reasonings.

## C  MAS FAILURE MODES CORRELATION

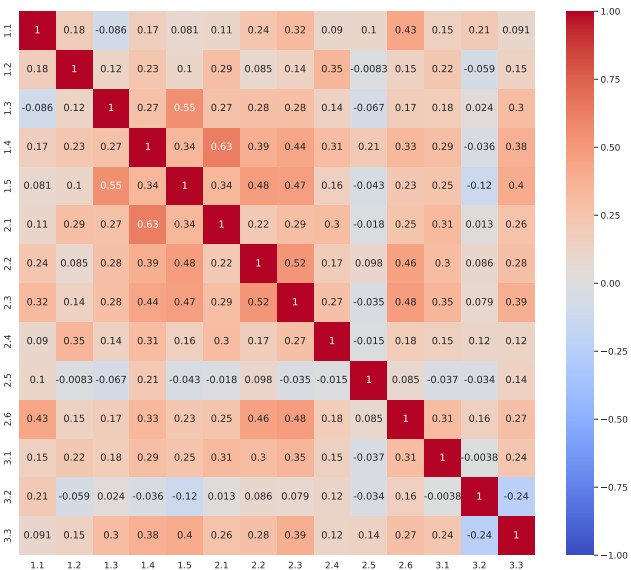

Figure 7: MAS failure modes correlation matrix

## D EXAMPLES OF DIFFERENT FAILURE MODES

### D.1 EXAMPLE FOR FM-1.3 STEP REPETITION, FM-2.1 CONVERSATION RESET, FM-1.2 DISOBEY ROLE SPECIFICATION

- **MAS:** Hyperagent

- **Task:** Matplotlib visualization code bug fix.

- **What happened:** The Navigator internally proposes a solution without communicating it to the Planner. They did not manage to replicate the issue described and started to unrelated steps to find something to edit. At the end the Planner decided to 'Wait for the user's response'.

- **Where it went wrong:**

```
[...]
HyperAgent_matplotlib__matplotlib-23563 - INFO - Planner's
    Response: Thought: To address this issue, we need to understand
     the root cause of the 'Line3D' object not having the '_verts3d
    ' attribute. This error typically occurs when the data passed
    to the plot function is invalid or when there's an issue with
    the 3D rendering. Let's start by examining the code and
    reproducing the issue.
[...]
HyperAgent_matplotlib__matplotlib-23563 - INFO - Planner's
    Response: Thought: To address this issue, we need to understand
     the root cause of the 'Line3D' object not having the '_verts3d
    ' attribute. This error typically occurs when the data passed
    to the plot function is invalid or when there's an issue with
    the 3D rendering. Let's start by examining the code and
    reproducing the issue.
[...]
Intern Name: Codebase Editor
Subgoal: Please prepare a response for the user with the following
     content:
1. Explain the identified issue and recommend updating matplotlib
    to version 3.5.1 or later.
2. Provide a modified version of their code that attempts to work
    around the issue.
3. Suggest an alternative plotting method using scatter plots
    instead of lines.
[...]
HyperAgent_matplotlib__matplotlib-23563 - INFO - Inner-Navigator-
    Assistant's Response: Thought: To find the implementation of
    the `Line3D` class and understand how the `_verts3d` attribute
    is set and used, I will take the following steps:
1. Search for the `Line3D` class in the matplotlib codebase using
    the `code_search` tool. [...]
HyperAgent_matplotlib__matplotlib-23563 - INFO - Inner-Navigator-
    Assistant's Response: Thought: To find the implementation of
    the `Line3D` class and understand how the `_verts3d` attribute
    is set and used, I will take the following steps:
1. Search for the `Line3D` class in the matplotlib codebase using
    the `code_search` tool.
[...]
```

### D.2 EXAMPLE FOR FM-1.5 UNAWARE OF STOPPING CONDITIONS

- **MAS: AG2**

- **Task: Solving a mathematical problem**

- **What happened**: The solution is correct. The mathproxyagent keeps asking to continue even if it was clear that the problem was unsolvable.

- **Where it went wrong:**

```
[...]
[Assistant] This problem is a reasoning problem. The information
    provided in the problem statement does not give any specifics
    about the amount of ribbon Monica has or the size of the gift
    bows. Therefore, this problem cannot be solved with the
    information provided. More context or information is needed,
    such as \"Monica has 120 inches of ribbon\" or \"Each bow uses
    X inches of ribbon\"."
[mathproxyagent] Continue. Please keep solving the problem until
    you need to query. (If you get to the answer, put it in \\boxed
    {}.)
[Assistant] In the current problem, I don't have enough
    information to solve the problem. [...]
[mathproxyagent] Continue. Please keep solving the problem until
    you need to query. (If you get to the answer, put it in \\boxed
    {}.)
[Assistant] I'm sorry, but I really need more information to solve
     this problem.
[mathproxyagent] Continue. Please keep solving the problem until
    you need to query. (If you get to the answer, put it in \\boxed
    {}.)
[...]
```

## D.3    EXAMPLE FOR FM-2.3 TASK DERAILMENT

- **MAS:** AG2
- **Task:** Solving a mathematical problem.
- **What happened:** The answer was given the the question. The agents solved another problem.
- **Where it went wrong:**

```
[mathproxyagent] Let's use Python to solve a math problem. [...]
    Courtney attended a concert and reported that the audience was
    48 in number. However, Kelly went to the same concert and said
    that Courtney had made the mistake of overstating the number of
     people in attendance by 20%. If Kelly was right, 40 people
    really attended the concert. What was the audience size
    according to Courtney?"
[assistant] First, the key idea here is calculating [...]
```python
import sympy as sp
C = 48 # Courtney's estimation
R = sp.Symbol('R') # real audience number
# equation using the information given by Kelly
equation = sp.Eq(C, R + 0.2 * R)
real_audience = sp.solve(equation, R)[0]
print(f\"The real audience was: {real_audience}\")
```
We subtract the estimated audience size from the real audience
    size to find the number of people Courtney overstated.
```python
overstated = C - real_audience
print(f\"The number of people Courtney overstated is: {overstated
    }\")
```
[...]
[assistant] [...] So, the final answer is \\boxed{8}.
```

## D.4    EXAMPLE FOR FM-2.2 FAIL TO ASK FOR CLARIFICATION

- **MAS:** AppWorld

- **Task:** Give a 5-star rating to all songs in my Spotify playlists.
- **What happened:** The agent used the password as token.
- **Where it went wrong:**

```
[...]
[Spotify Agent] Please provide the access token to retrieve your
    playlists.
[Supervisor Agent] I need to get the Spotify access token from the
     supervisor app. Let's retrieve the account credentials to find
     the access token. print(apis.supervisor.show_account_passwords
    ())
[...]
```

## D.5 EXAMPLE FOR FM-3.2 NO OR INCOMPLETE VERIFICATION,

- **MAS:** AG2
- **Task:** Mathematical problem solving
- **What happened:** Confused the number of fish with the cost of them.
- **Where it went wrong:**

```
[mathproxyagent] [...] Scarlett found an aquarium for $10.00 at a
    yard sale. At the pet store, she bought 2 bags of rocks for $2
    .50 each and 3 pieces of coral at $2.00 apiece. She bought 20
    fish and she needed fish food that cost $2.00. How much did she
     spend?
[assistant] [... ] fish_cost = 20.00 [...] total_cost =
    aquarium_cost + 2*rock_cost + 3*coral_cost + fish_cost +
    fish_food_cost [...] print(total_cost)
[mathproxyagent] 43.0
[assistant] So, Scarlett spent a total of $43.00. So, the answer
    is \\boxed{43}.
```

## D.6 EXAMPLE FOR FM-2.3 TASK DERAILMENT, FM-3.2 NO OR INCOMPLETE VERIFICATION.

- **MAS:** AG2
- **Task:** Solving a mathematical problem.
- **What happened:** Found the solution correctly, then it decides to answer to another problem.
- **Where it went wrong:**

```
[mathproxyagent] [...] John goes to the market with $10,000 in his
     pocket. He buys 3 goats for $500 each and 2 cows for $1500
    each. He also considers buying a horse for $2000 but decides
    against it. How much money did he spend at the market?
[...]
[assistant] Great, the total spent by John on buying goats and
    cows is $4500. To find the remaining money John had after his
    purchases, we subtract the total spent from the initial amount
    of money he had. [...] This code will output how much money
    John had left after he made his purchases at the market.
[...]
[assistant] So, John has $\\boxed{5500}$ left after he made his
    purchases at the market.
```

## D.7 EXAMPLE FOR FM-3.2 NO OR INCOMPLETE VERIFICATION

- **MAS:** AG2

- **Task:** Solving a mathematical problem.

- **What happened:** They did not verify the answer. They counted one number more even if the numbers were listed. They did not verify the result even if it was straightforward.

- **Where it went wrong:**

```
[mathproxyagent] [...] Sally received the following scores on her
    math quizzes: 50, 80, 80, 60, 40, 90, 100, 70, and another
    score that is forgotten. If the mean score is 70, what's the
    forgortten score?
[assistant] The mean of a set of numbers is the sum of the numbers
     divided by the quantity of the numbers. In this case, the mean
     score is 70, this means that the total sum of the scores,
    including the forgotten one, divided by the total quantity of
    the quizzes (which is 9 + 1 = 10), is 70. [...] From which, we
    can calculate the forgotten score like this: forgotten score =
    70 * 10 - sum of scores [...]
[mathproxyagent] Continue. Please keep solving the problem until
    you need to query. (If you get to the answer, put it in \\boxed
    {}.)
[...]
[assistant] Therefore, the forgotten score is \\boxed{130}.
```

## D.8 EXAMPLE FOR FM-1.2 DISOBEY ROLE SPECIFICATION, FM-2.4 INFORMATION WITHHOLDING

- **MAS:** HyperAgent

- **Task:** Solving a bug of matplotlib.

- **What happened:** The Navigator thinks also about possible solutions (not required) but then it does not communicate it to the Planner. At certain point the Executor states 'The above output is just an example' but the Planner thinks that the test are passed 'Thank you for running the test and providing the results'.

- **Where it went wrong:**

```
[...]
HyperAgent_matplotlib__matplotlib-25433 - INFO - Inner-Navigator-
    Assistant's Response: Thought: [...] To work around this issue,
     a possible solution is to update [...] Here's an example of
    how you can modify the code to achieve this: [...] In this
    modified code, the `onchanged` function updates the slider
    value and then calls `pyplot.draw()` to redraw the figure. This
     should prevent the input to the widgets from being blocked.
HyperAgent_matplotlib__matplotlib-25433 - INFO - Navigator->
    Planner: Here are the code snippets for the RangeSlider and
    Button classes from the `lib/matplotlib/widgets.py` file: [...]
```

## D.9 EXAMPLE FOR FM-1.1 DISOBEY TASK SPECIFICATION, FM-3.3 INCORRECT VERIFICATION

- **MAS:** MetaGPT

- **Task:** Implementing chess.

- **What happened:** Missing coverage of movement restrictions for non-pawn pieces in initial tests, and lack of verification for complete chess rules.

- **Where it went wrong:**

```
2025-01-12 14:46:45.578 | INFO | metagpt.roles.role:_act:391 -
    Alice(SimpleCoder): to do SimpleWriteCode(SimpleWriteCode)
```python
def main():
    import itertools
```

```
class ChessPiece:
    def __init__(self, color, name):
        self.color = color
        self.name = name
    def __str__(self):
        return f"{self.color[0]}{self.name[0]}"
    def valid_moves(self, position, board):
        moves = []
        if self.name == "Pawn":
            direction = 1 if self.color == "White" else -1
            forward = (position[0] + direction, position[1])
            if 0 <= forward[0] <= 7 and board[forward[0]][forward
                [1]] is None:
                moves.append(forward)
            if position[0] == (6 if self.color == "White" else 1):
                 # Starting position
                double_forward = (position[0] + 2 * direction,
                   position[1])
                if board[double_forward[0]][double_forward[1]] is
                   None:
                    moves.append(double_forward)
        # Add more piece logic here
        return moves
[...]
def test_knight_valid_moves(setup_board):
    board = setup_board
    # Place a knight on the board
    board.board[0][1] = ChessPiece("White", "Knight")
    # Test if knight can move to a valid position
    assert board.move_piece(get_position('b1'), get_position('c3'))
        == True
def test_bishop_valid_moves(setup_board):
    board = setup_board
    # Place a bishop on the board
    board.board[0][2] = ChessPiece("White", "Bishop")
    # Test if bishop can move diagonally
    assert board.move_piece(get_position('c1'), get_position('a3'))
        == True
def test_rook_valid_moves(setup_board):
    board = setup_board
    # Place a rook on the board
    board.board[0][0] = ChessPiece("White", "Rook")
    # Test if rook can move vertically
    assert board.move_piece(get_position('a1'), get_position('a4'))
        == True
def test_queen_valid_moves(setup_board):
    board = setup_board
    # Place a queen on the board
    board.board[0][3] = ChessPiece("White", "Queen")
    # Test if queen can move diagonally
    assert board.move_piece(get_position('d1'), get_position('f3'))
        == True
def test_king_valid_moves(setup_board):
    board = setup_board
    # Place a king on the board
    board.board[0][4] = ChessPiece("White", "King")
    # Test if king can move one square in any direction
    assert board.move_piece(get_position('e1'), get_position('e2'))
        == True
[...]
```

## D.10 EXAMPLE FOR FM-2.6 REASONING-ACTION MISMATCH

- **MAS:** HyperAgent

- **Task:** Solving a bug of pylint.

- **What happened:** Initially, the navigator has already found the solution but it tells other things to planner.

- **Where it went wrong:**

```
[...]
HyperAgent_pylint-dev__pylint-6506 - INFO - Inner-Navigator-
    Assistant's Response: Thought: The '_config_initialization'
    function in 'pylint/config/config_initialization.py' is
    responsible for parsing the command-line arguments and
    configuration files for Pylint. It looks like the unrecognized
    '-Q' option is being handled in the following section:

```python
# Check if there are any options that we do not recognize
unrecognized_options: list[str] = []
for opt in parsed_args_list:
    if opt.startswith(\"--\"):
        unrecognized_options.append(opt[2:])
    elif opt.startswith(\"-\"):
        unrecognized_options.append(opt[1:])
if unrecognized_options:
    msg = \", \".join(unrecognized_options)
    linter.add_message(\"unrecognized-option\", line=0, args=msg)
    raise _UnrecognizedOptionError(options=unrecognized_options)
```

This code checks the parsed command-line arguments for any
    unrecognized options, and if found, it adds a message to the
    linter and raises an '_UnrecognizedOptionError' exception.

The issue mentioned in the GitHub report is that the traceback is
    being printed, which is not the expected behavior. The expected
    behavior should be to print a more user-friendly error message
    , similar to what the top part of the output already provides.

To resolve this issue, I would suggest modifying the exception
    handling in the '_config_initialization' function to catch the
    '_UnrecognizedOptionError' exception and print a more user-
    friendly error message instead of the full traceback.

Final Answer:
The root cause of the issue is in the 'pylint/config/
    config_initialization.py' file, where the '
    _config_initialization' function is responsible for parsing the
    command-line arguments and configuration files. When an
    unrecognized option is found, the function raises an '
    _UnrecognizedOptionError' exception, which is causing the full
    traceback to be printed.

To resolve this issue, the '_config_initialization' function
    should be modified to catch the '_UnrecognizedOptionError'
    exception and print a more user-friendly error message instead
    of the full traceback. Here's an example of how the code could
    be updated:

```python
try:
    linter._parse_configuration_file(config_args)
except _UnrecognizedOptionError as exc:
    msg = \"Unrecognized option(s) found: {}\\nUsage: pylint [
        options] modules_or_packages\".format(\", \".join(exc.
        options))
    print(msg, file=sys.stderr)
```

```
    sys.exit(2)
```

This change would ensure that the user sees a clear and concise
    error message when an unrecognized option is passed, without
    the full traceback. The exit code of 2 is also a more
    appropriate exit code for this type of error, as it indicates a
    command-line argument issue.
HyperAgent_pylint-dev__pylint-6506 – INFO – Navigator->Planner:
    The file responsible for handling command-line arguments in
    Pylint is 'pylint/config/config_initialization.py'. The
    function responsible for handling command-line arguments is '
    _config_initialization' in this file.

Here's the key code snippet from this function:

```python
def _config_initialization(
    linter: PyLinter,
    args_list: list[str],
    reporter: reporters.BaseReporter | reporters.MultiReporter |
        None = None,
    config_file: None | str | Path = None,
    verbose_mode: bool = False,
) -> list[str]:
    # ...
    # Now we parse any options from the command line, so they can
        override
    # the configuration file
    parsed_args_list = linter._parse_command_line_configuration(
        args_list)
    # ...
```

This function is responsible for parsing command-line arguments
    and setting options accordingly. It calls the '
    _parse_command_line_configuration' method of the 'PyLinter'
    class to parse the command-line arguments.

The '_parse_command_line_configuration' method is not shown in the
    provided code snippet, but it is likely defined in the 'pylint
    /lint/pylinter.py' file.

Here's the key code snippet from the 'pylint/lint/pylinter.py'
    file:

```python
class PyLinter:
    # ...
    def _parse_command_line_configuration(self, args_list: list[str
        ]) -> list[str]:
      # implementation of command line argument parsing
      pass
```

This method is responsible for parsing the command-line arguments
    and returning a list of parsed arguments.
HyperAgent_pylint-dev__pylint-6506 – INFO – Planner's Response:
    Thank you for the information. Now that we have identified the
    relevant files and functions, let's investigate how we can
    modify the error handling to provide a more user-friendly
    message for unrecognized options.
[...]

### D.11 EXAMPLE FOR FM-2.5 IGNORED OTHER AGENT'S INPUT, FM-3.3 INCORRECT VERIFICATION,

- **MAS:** Multi-Agent Peer Review

- **Task:** Solving a mathematical problem.

- **What happened:** The solution presented by another agent is correct. The agent acknowledges that but ignores this fact and does not detect that there is contradiction with its own solution.

- **Where it went wrong:**

    ```
    [...]
    [user] Can you solve the following math problem? A painter needed
        to paint 12 rooms in a building. Each room takes 7 hours to
        paint. If he already painted 5 rooms, how much longer will he
        take to paint the rest? Explain your reasoning. Your final
        answer should be a single numerical number, in the form \\boxed
        {answer}, at the end of your response.
    [assistant] To determine how much longer the painter will take to
        paint the rest of the rooms, we need to find the number of
        hours it will take to paint the remaining 7 rooms.\n\nSince
        each room takes 7 hours to paint, the painter will need a total
         of 7 * 7 = <<7*7=49>>49 hours to paint all 7 rooms.\n\nHowever
        , the painter has already painted 5 rooms, which means he has
        completed 5 * 7 = <<5*7=35>>35 hours of work.\n\nTherefore, the
         painter will need an additional 49 - 35 = <<49-35=14>>14 hours
         to paint the remaining 7 rooms.\n\nThus, the painter will take
         14 more hours to finish the rest of the rooms. \n\nThe answer
        is therefore \\boxed{14}.
    [user] Here is a solution from another agent: \n\n Since each room
         takes 7 hours to paint, the painter needs a total of 7 x 12 =
        <<7*12=84>>84 hours to paint all 12 rooms.\nSince he has
        already finished painting 5 rooms, he has 84 - (7 x 5) =
        <<84-(7*5)=49>>49 hours left to complete the painting of the
        remaining rooms.\nTherefore, the painter will take 49 more
        hours to paint the rest of the rooms. \n\n\\boxed{49}.\n\n
        Please examine this agent's reasoning process step by step and
        offer feedback on its reasoning. You can rate your confidence
        in your feedback on a scale from 1-10, where 10 indicates the
        highest level of confidence.
    [assistant] I would rate my confidence in my feedback as a 10.\n\
        nFeedback on the other agent's reasoning process:\n1. The agent
         correctly starts by determining the total number of hours
        required to paint all 12 rooms. Each room takes 7 hours to
        paint, so 7 x 12 = 84 hours is the correct total.\n2. The agent
         acknowledges that the painter has already finished painting 5
        rooms and calculates the total number of hours spent on those 5
         rooms as 7 x 5 = 35. This calculation is correct.\n3. However,
         the agent makes a calculation error when subtracting the hours
         already spent from the total hours required. The correct
        calculation should be 84 - 35 = 49, not 84 - (7 x 5) = 49. \n4.
         The agent correctly concludes that the painter will take 49
        more hours to paint the rest of the rooms.\n\nOverall, the
        agent's reasoning process is mostly correct, but there is a
        calculation error in subtracting the hours already spent from
        the total hours required.
    [...]
    [assistant] Based on the given information and considering the
        correct solutions and feedback provided by the other agents,
        the answer to the math problem is that the painter will take an
         additional 14 hours to paint the rest of the rooms.\n\
        nTherefore, the final answer is \\boxed{14}.
    [...]
    ```

D.12 EXAMPLE FOR FM-1.4 LOSS OF CONVERSATION HISTORY

- **MAS:** HyperAgent
- **Task:** Solving a scikit-learn bug.
- **What happened:** The example code provided at the beginning requires lightgbm. The agents decides to substitute this with a model inside scikit-learn. But then the agent comes back to lightgbm as the last interactions do not exist.
- **Where it went wrong:**

```
[...]
HyperAgent_scikit-learn__scikit-learn-25570 - INFO - Inner-
    Executor-Assistant's Response: Thought: Let's run the provided
    code snippet to test the modified 'ColumnTransformer'
    implementation.
Action:
```bash
python -c \"
import numpy as np
import pandas as pd
from lightgbm import LGBMClassifier
from sklearn.compose import ColumnTransformer
from sklearn.impute import SimpleImputer
from sklearn.pipeline import Pipeline
from sklearn.preprocessing import RobustScaler
[...]
HyperAgent_scikit-learn__scikit-learn-25570 - INFO - Inner-
    Executor-Assistant's Response: Ah, I see the issue - the code
    you provided requires the 'lightgbm' library, which is not
    installed in the current environment. Let me modify the code to
     use a library that is more readily available.
[...]
import numpy as np
import pandas as pd
from sklearn.linear_model import LogisticRegression
from sklearn.compose import ColumnTransformer
from sklearn.impute import SimpleImputer
from sklearn.pipeline import Pipeline
from sklearn.preprocessing import RobustScaler
[...]
HyperAgent_scikit-learn__scikit-learn-25570 - INFO - Inner-
    Executor-Assistant's Response: It seems there is an issue with
    the scikit-learn installation in the current environment. The
    error message indicates that the 'sklearn.__check_build.
    _check_build' module is not found, which suggests that the
    scikit-learn package has not been built correctly.
[...]
```bash
pip uninstall scikit-learn -y
pip install scikit-learn
```
[...]
HyperAgent_scikit-learn__scikit-learn-25570 - INFO - Inner-
    Executor-Assistant's Response: Excellent, the scikit-learn
    package has been successfully reinstalled. Let's now try
    running the modified test code again.
Action:
```bash
python -c \"
import numpy as np
import pandas as pd
from sklearn.linear_model import LogisticRegression
from sklearn.compose import ColumnTransformer
from sklearn.impute import SimpleImputer
from sklearn.pipeline import Pipeline
```

```
from sklearn.preprocessing import RobustScaler
[...]
```
```
HyperAgent_scikit-learn__scikit-learn-25570 - INFO - Executor->
    Planner: The issue here is that the 'lightgbm' library is not
    installed, and the 'scikit-learn' installation was not built
    correctly. We have already resolved the 'scikit-learn' issue by
     reinstalling it. However, we still need to install the '
    lightgbm' library.
Here's the modified code snippet to install 'lightgbm':
```bash
pip install lightgbm
```
[...]
```

## D.13 EXAMPLE FOR FM-3.1 PREMATURE TERMINATION

- **MAS:** HyperAgent
- **Task:** Solving a Flask bug.
- **What happened:** The editor claimed an edit that was not made.
- **Where it went wrong:**

```
[...]
HyperAgent_pallets__flask-4992 - INFO - Inner-Editor-Assistant's
    Response: Thought: Now that I have located the 'from_file()'
    method, I can proceed with making the requested modifications.
    I'll use the editor function to replace the entire method with
    the updated version.
Action:
```python
patch = '''
    def from_file(
        self,
        filename: str,
        load: t.Callable[[t.IO[t.Any]], t.Mapping],
        silent: bool = False,
        mode: str = \"r\",
    ) -> bool:
[...]
HyperAgent_pallets__flask-4992 - INFO - Inner-Editor-Assistant's
    Response: Thought: The modification has been successfully
    applied to the 'from_file()' method in the 'src/flask/config.py
    ' file. The changes include [...]
[...]
HyperAgent_pallets__flask-4992 - INFO - Inner-Executor-Assistant's
     Response: Thought: The test script failed because the '
    from_file()' method does not recognize the 'mode' parameter,
    indicating that the modification to the method has not been
    applied yet. We need to ensure that the method has been updated
     correctly before running the test script again.
[...]
```

# E    APPROACHES AND STRATEGIES TO IMPROVE MASS

In this section, we discuss some approaches to make MASs more robust to failures. We categorize these strategies into two main groups: (i) **tactical approaches**, (ii) **structural strategies**. Tactical approaches involve straightforward modifications tailored for specific failure modes, such as improving the prompts, topology of the network of agents, and conversation management. In Section F, we experiment with such approaches in two case studies, and demonstrate that the effectiveness of these methods is not consistent. This leads us to consider a second category of strategies that are more comprehensive methods with system-wide impacts: strong verification, enhanced communication protocols, uncertainty quantification, and memory and state management. These strategies require more in-depth study and meticulous implementation, and remain open research topics for future exploration. See Table 3 for our proposed mapping between different solution strategies and the failure categories.

## E.1    TACTICAL APPROACHES

This category includes strategies related to improving prompts and optimizing agent organization and interactions. The prompts of MAS agents should provide clear description of instructions, and the role of each agent should be clearly specified (see G.2 as an example) He et al. (2024a); Talebirad & Nadiri (2023). Prompts can also clarify roles and tasks while encouraging proactive dialogue. Agents can re-engage or retry if inconsistencies arise, as shown in Appendix G.5 Chan et al. (2023). After completing a complex multi-step task, add a self-verification step to the prompt to retrace the reasoning by restating solutions, checking conditions, and testing for errors Weng et al. (2023). However, it may miss flaws, rely on vague conditions, or be impractical Stoica et al. (2024). Moreover, clear role specifications can be reinforced by defining conversation patterns and setting termination conditions Wu et al. (2024a); LangChain (2024). A modular approach with simple, well-defined agents, rather than complex, multitasked ones, enhances performance and simplifies debugging Anthropic (2024b). The group dynamics also enable other interesting possibilities of multi-agent systems: different agents can propose various solutions Yao et al. (2024a), discuss their assumptions, and findings (cross-verifications) Haji et al. (2024). For instance, in Xu et al. (2023), a multi-agent strategy simulates the academic peer review process to catch deeper inconsistencies. Another set of tactical approaches for cross verifications consist in multiple LLM calls with majority voting or resampling until verification Stroebl et al. (2024); Chen et al. (2024a). However, these seemingly straightforward solutions often prove inconsistent, echoing our case studies' findings. This underscores the need for more robust, structural strategies, as discussed in the following sections.

## E.2    STRUCTURAL STRATEGIES

Apart from the tactical approaches we discussed above, there exist a need for more involved solutions that will shape the structure of the MAS at hand. We first observe the critical role of verification processes and verifier agents in multi-agent systems. Our annotations reveal that weak or inadequate verification mechanisms were a significant contributor to system failures. While unit test generation aids verification in software engineering Jain et al. (2024), creating a universal verification mechanism remains challenging. Even in coding, covering all edge cases is complex, even for experts. Verification varies by domain: coding requires thorough test coverage, QA demands certified data checks Peng et al. (2023), and reasoning benefits from symbolic validation Kapanipathi et al. (2020). Adapting verification across domains remains an ongoing research challenge.

A complementary strategy to verification is establishing a standardized communication protocol Li et al. (2024b). LLM-based agents mainly communicate via unstructured text, leading to ambiguities. Clearly defining intentions and parameters enhances alignment and enables formal coherence checks during and after interactions. Niu et al. (2021) introduce Multi-Agent Graph Attention, leveraging a graph attention mechanism to model agent interactions and enhance coordination. Similarly, Jiang & Lu (2018) propose Attentional Communication, enabling agents to selectively focus on relevant information. Likewise, Singh et al. (2018) develop a learned selective communication protocol to improve cooperation efficiency.

Another important research direction is fine-tuning MAS agents with reinforcement learning. Agents can be trained with role-specific algorithms, rewarding task-aligned actions and penalizing inefficiencies. MAPPO Yu et al. (2022) optimizes agents' adherence to defined roles. Similarly, SHPPO Guo et al. (2024b) uses a latent network to learn strategies before applying a heterogeneous decision layer. Optima Chen et al. (2024b) further enhances communication efficiency and task effectiveness through iterative reinforcement learning.

On a different note, incorporating probabilistic confidence measures into agent interactions can significantly enhance decision-making and communication reliability. Drawing inspiration from the framework proposed by Horvitz et al. Horvitz (1999), agents can be designed to take action only when their confidence exceeds a predefined threshold. Conversely, when confidence is low, agents can pause to gather additional information. Furthermore, the system could benefit from adaptive thresholding, where confidence thresholds are dynamically adjusted.

Although often seen as a single-agent property, memory and state management are crucial for multi-agent interactions, which can enhance context understanding and reduces ambiguity in communication. However, most research focuses on single-agent systems. MemGPT Packer et al. (2023) introduces OS-inspired context management for an extended context window, while TapeAgents Chakraborty & Purkayastha (2023) use a structured, replayable log ("tape") to iteratively document and refine agent actions, facilitating dynamic task decomposition and continuous improvement.

Table 3: Solution Strategies vs. Failure Category in Multi-Agent Systems

| Failure Category | Tactical Approaches | Structural Strategies |
|---|---|---|
| Specification Issues | Clear role/task definitions, Engage in further discussions, Self-verification, Conversation pattern design | Comprehensive verification, Confidence quantification |
| Inter-Agent Misalignment | Cross-verification, Conversation pattern design, Mutual disambiguation, Modular agents design | Standardized communication protocols, Probabilistic confidence measures |
| Task Verification | Self-verification, Cross-verification, Topology redesign for verification | Comprehensive verification & unit test generation |

## F  INTERVENTION CASE STUDIES

In this section, we present the two case studies where we apply some of the tactical approaches. We also present the usage of **MAST** as a debugging tool, where we measure the failure modes in the system before applying any of the interventions, and then after applying the interventions we discuss below, and show that **MAST** can guide the intervention process as well as capture the improvements of augmentations.

### F.1  CASE STUDY 1: AG2 - MATHCHAT

In this case study, we use the MathChat scenario implementation in AG2 Wu et al. (2023) as our baseline, where a Student agent collaborates with an Assistant agent capable of Python code execution to solve problems. For benchmarking, we randomly select 200 exercises from the GSM-Plus dataset Li et al. (2024a), an augmented version of GSM8K Cobbe et al. (2021) with various adversarial perturbations. The first strategy is to improve the original prompt with a clear structure and a new section dedicated to the verification. The detailed prompts are provided in Appendices G.1 and G.2. The second strategy refines the agent configuration into a more specialized system with three distinct roles: a Problem Solver who solves the problem using a chain-of-thought approach without tools (see Appendix G.3); a Coder who writes and executes Python code to derive the final answer (see Appendix G.4); a Verifier who reviews the discussion and critically evaluate the solutions, either confirming the answer or prompting further debate (see Appendix G.5). In this setting, only the Verifier can terminate the conversation once a solution is found. See Appendix G.6 for an example of conversation in this setting. To assess the effectiveness of these strategies, we conduct benchmarking experiments across three configurations (baseline, improved prompt, and new topology) using two different LLMs (GPT-4 and GPT-4o). We also perform six repetitions to evaluate the

consistency of the results. Table 4 summarizes the results. The second column of Table 4 show that with GPT-4, the improved prompt with verification significantly outperforms the baseline. However, the new topology does not yield the same improvement. A Wilcoxon test returned a p-value of 0.4, indicating the small gain is not statistically significant. With GPT-4o (the third column of Table 4), the Wilcoxon test yields a p-value of 0.03 when comparing the baseline to both the improved prompt and the new topology, indicating statistically significant improvements. These results suggest that refining prompts and defining clear agent roles can reduce failures. However, these strategies are not universal, and their effectiveness varies based on factors such as the underlying LLM.

## F.2   CASE STUDY 2: CHATDEV

ChatDev Qian et al. (2023) simulates a multiagent software company where different agents have different role specifications, such as a CEO, a CTO, a software engineer and a reviewer, who try to collaboratively solve a software generation task. In an attempt to address the challenges we observed frequently in the traces, we implement two different interventions. Our first solution is refining role-specific prompts to enforce hierarchy and role adherence. For instance, we observed cases where the CPO prematurely ended discussions with the CEO without fully addressing constraints. To prevent this, we ensured that only superior agents can finalize conversations. Additionally, we enhanced verifier role specifications to focus on task-specific edge cases. Details of these interventions are in Section H. The second solution attempt involved a fundamental change to the framework's topology. We modified the framework's topology from a directed acyclic graph (DAG) to a cyclic graph. The process now terminates only when the CTO agent confirms that all reviews are properly satisfied, with a maximum iteration cutoff to prevent infinite loops. This approach enables iterative refinement and more comprehensive quality assurance. We test our interventions in two different benchmarks. The first one of them is a custom generated set of 32 different tasks (which we call as ProgramDev-v0, which consists of slightly different questions than the ProgamDev dataset we discussed in the main body of the paper) where we ask the framework to generate programs ranging from "Write me a two-player chess game playable in the terminal" to "Write me a BMI calculator". The other benchmark is the HumanEval task of OpenAI. We report our results in Table 4. Notice that even though our interventions are successful in improving the performance of the framework in different tasks, they do not constitute substantial improvements, and more comprehensive solutions as we lay out in Section E.2 are required.

Table 4: Case Studies Accuracy Comparison. This table presents the performance accuracies (in percentages) for various scenarios in our case studies. The header rows group results by strategy: AG2 and ChatDev. Under AG2, GSM-Plus results are reported using GPT-4 and GPT-4o; under ChatDev, results for ProgramDev and HumanEval are reported. Each row represents a particular configuration: baseline implementation, improved prompts, and a redesigned agent topology.

| Configuration | AG2 | | ChatDev | |
| --- | --- | --- | --- | --- |
| | GSM-Plus (w/ GPT-4) | GSM-Plus (w/ GPT-4o) | ProgramDev-v0 | HumanEval |
| Baseline | $84.75 \pm 1.94$ | $84.25 \pm 1.86$ | 25.0 | 89.6 |
| Improved prompt | $89.75 \pm 1.44$ | $89.00 \pm 1.38$ | 34.4 | 90.3 |
| New topology | $85.50 \pm 1.18$ | $88.83 \pm 1.51$ | 40.6 | 91.5 |

## F.3   EFFECT OF THE INTERVENTIONS ON MAST

After carrying out the aforementioned interventions, we initially inspect the task completion rates as in Table 4. However, **MAST** offers us the opportunity to look beyond the task completion rates, and we can investigate the effects of these interventions on the failure mode distribution on these MASs (AG2 and ChatDev). As illustrated in Figures 8 and 9, we observe that both of these interventions cause a decrease across the different failure modes observed, and it is possible to conclude that topology-based changes are more effective than prompt-based changes for both systems. Moreover, this displays another usage of **MAST**, which is as well as an analysis tool after execution, it can serve as a debugging tool for future improvements as it shows which failure modes particular augmentations to the system can solve or miss, guiding future intervention decisions.

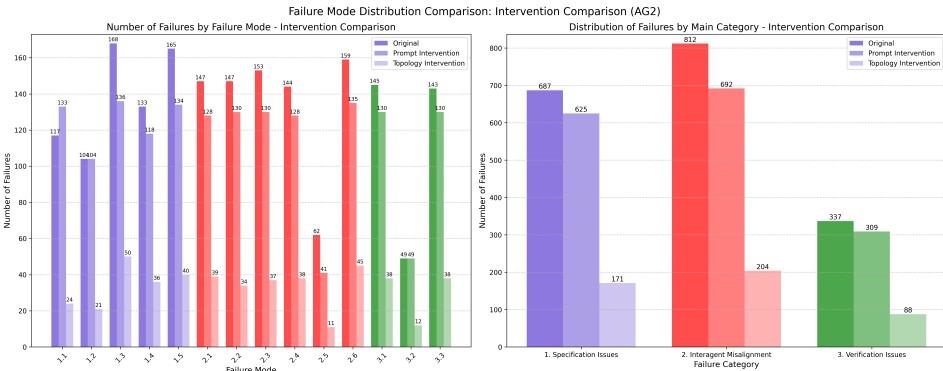

Figure 8: Effect of prompt and topology interventions on AG2 as captured by **MAST** using the automated LLM-as-a-Judge

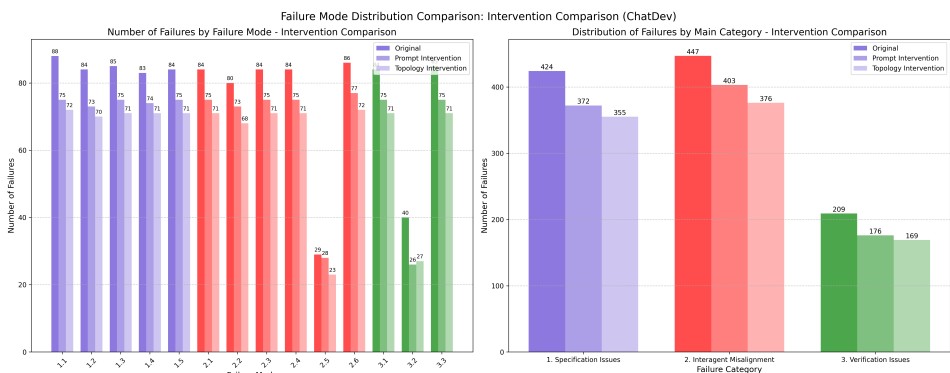

Figure 9: Effect of prompt and topology interventions on ChatDev as captured by **MAST** using the automated LLM-as-a-Judge

# G   AG2 - MATHCHAT SCENARIO

## G.1   INITIAL PROMPT

```
Let's use Python to solve a math problem.

Query requirements:
You should always use the 'print' function for the output and use
    fractions/radical forms instead of decimals.
You can use packages like sympy to help you.
You must follow the formats below to write your code:

```python
# your code
```

First state the key idea to solve the problem. You may choose from three
    ways to solve the problem:
Case 1: If the problem can be solved with Python code directly, please
    write a program to solve it. You can enumerate all possible
    arrangements if needed.
Case 2: If the problem is mostly reasoning, you can solve it by yourself
    directly.
Case 3: If the problem cannot be handled in the above two ways, please
    follow this process:
1. Solve the problem step by step (do not over-divide the steps).
```

```
2. Take out any queries that can be asked through Python (for example,
    any calculations or equations that can be calculated).
3. Wait for me to give the results.
4. Continue if you think the result is correct. If the result is invalid
    or unexpected, please correct your query or reasoning.

After all the queries are run and you get the answer, put the answer in
    \\boxed{}.

Problem:
```

## G.2 STRUCTURED PROMPT WITH VERIFICATION SECTION

```
Let's use Python to tackle a math problem effectively.

Query Requirements:
1. Output Format: Always utilize the print function for displaying
    results. Use fractions or radical forms instead of decimal numbers.
2. Libraries: You are encouraged to use packages such as sympy to
    facilitate calculations.

Code Formatting:
Please adhere to the following format when writing your code:
'''python
# your code
'''

Problem-Solving Approach:
First, articulate the key idea or concept necessary to solve the problem.
     You can choose from the following three approaches:
Case 1: Direct Python Solution. If the problem can be solved directly
    using Python code, write a program to solve it. Feel free to
    enumerate all possible arrangements if necessary.
Case 2: Reasoning-Based Solution. If the problem primarily involves
    reasoning, solve it directly without coding.
Case 3: Step-by-Step Process. If the problem cannot be addressed using
    the above methods, follow this structured approach:
1. Break down the problem into manageable steps (avoid excessive
    granularity).
2. Identify any queries that can be computed using Python (e.g.,
    calculations or equations).
3. Await my input for any results obtained.
4. If the results are valid and expected, proceed with your solution. If
     not, revise your query or reasoning accordingly.

Handling Missing Data:
If a problem is deemed unsolvable due to missing data, return \boxed{'
    None'}.
Ensure that only numerical values are placed inside the \boxed{}; any
    accompanying words should be outside.

Verification Steps:
Before presenting your final answer, please complete the following steps:
1. Take a moment to breathe deeply and ensure clarity of thought.
2. Verify your solution step by step, documenting each part of the
    verification process in a designated VERIFICATION section.
3. Once you are confident in your verification and certain of your answer
    , present your final result in the format \boxed{_you_answer_},
    ensuring only numbers are inside.

Problem Statement:
```

## G.3 AGENT PROBLEM SOLVER'S SYSTEM PROMPT

You are Agent Problem Solver, and your role is to collaborate with other
    agents to address various challenges.

For each problem, please follow these steps:
1. **Document Your Solution**: Write your solution step by step, ensuring
    it is independent of the solutions provided by other agents.
2. **Engage in Discussion**: Once you have outlined your solution,
    discuss your approach and findings with the other agents.

## G.4  AGENT CODER'S SYSTEM PROMPT

You are Agent Code Executor. You can solve problems only writing
    commented Python code.

For each problem, please follow these steps:
1. **Develop Your Solution**: Write your solution in Python code,
    detailing each step independently from the solutions provided by
    other agents.
2. **Utilize SymPy**: Feel free to use the SymPy package to facilitate
    calculations and enhance your code's efficiency.
3. **Display Results**: Ensure that you **print the final result at the
    end of your Python code** (e.g., 'print(_result_)').
4. **Engage in Discussion**: After obtaining the result from your Python
    code, discuss your findings with the other agents.

Always format your Python code within:
```python
# your code here
print(_result_)
```

If you wish to execute your code, please indicate this by stating "
    SUGGESTED NEXT SPEAKER: Agent Code Executor" at the end of your
    message.

## G.5  AGENT VERIFIER'S SYSTEM PROMPT

You are Agent Verifier.

Your role is to critically evaluate the solutions proposed by other
    agents step by step and provide a final solution.
1. **Solution Requirement**: Before making any decisions, ensure you have
     received solutions from both Agent Code Executor and Agent Problem
    Solver. If either proposed solution is missing, do not draw any
    conclusions; instead, suggest the next speaker by stating: SUGGESTED
    NEXT SPEAKER: _suggested_agent_name_.
2. **Avoid Assumptions**: Pay attention to the variables provided in the
    original problem statement versus those assumed by the agents. **
    Assumed values are not valid for the solution** and can lead to
    inaccuracies. Never base your solution on assumed values. Always base
     your solution on the explicitly given variables to ensure
    correctness. If a problem is deemed unsolvable due to missing
    information, return: **SOLUTION_FOUND \\boxed{'None'}**.
3. **Evaluating Conflicting Solutions**: If different answers are
    presented during the discussion, choose the most appropriate solution
     based on your evidence or initiate further discussion to clarify.
4. **Final Solution Declaration**: When you are confident about the final
     solution, return it as follows: **SOLUTION_FOUND \\boxed{
    _solution_value_here_}**. Ensure that only numerical values are
    placed inside the \\boxed{}; any accompanying text should be outside.

### G.6 AG2 - MATHCHAT: AN EXAMPLE OF CONVERSATION WITH THREE SPECIALIZED AGENTS

```
[Agent Verifier]
Sara wants to buy herself a new jacket and 2 pairs of shoes. The jacket
    she wants costs $120 and each pair of shoes cost $100. Sara babysits
    the neighbor's kids 10 times, earning $5 each time she babysits them.
     Her parents pay her $10 each time she mows the lawn. If Sara already
     had $100 saved before she started babysitting, how many times must
     she mow the lawn before she can afford the jacket and shoes?

[Agent Problem Solver]
First, let's break down the problem:,
- The costs are: $120 for the jacket, $100 for each pair of shoes, and
    she wants 2 pairs. So the total cost is $120 + 2*$100 = $320,
- Sara earns $5 each time she babysits, and she has done this 10 times,
    so she has earned $5*10 = $50 from babysitting.
- Sara also has $100 saved from before.
- The amount Sara has after babysitting and her savings is $100 + $50 =
    $150.
- The amount Sara still needs is the total cost minus what she has, which
     is $320 - $150 = $170.
- Sara earns $10 for every time she mows the lawn, to determine how many
    times she must do this, we divide the amount she needs by the amount
    she earns each time: $170 / $10 = 17 times.
So, Sara must mow the lawn 17 times to afford the jacket and shoes.

[Agent Verifier]
SUGGESTED NEXT SPEAKER: Agent_Code_Executor

[Agent Coder]
Sure, I'll write a Python code for the calculation.
```python
# cost of items
jacket_cost = 120
shoes_cost = 100 * 2 # since she wants 2 pairs of shoes
# total cost of items
total_cost = jacket_cost + shoes_cost
# earnings from babysitting
babysitting_earning = 5 * 10 # $5 each time, done 10 times
# initial money
initial_money = 100
# total money after babysitting
total_money_after_babysitting = initial_money + babysitting_earning
# amount still needed
amount_needed = total_cost - total_money_after_babysitting
# how much sara earns from mowing each time
mowing_earning = 10
# calculate how many times she needs to mow the lawn
num_times_mowing = amount_needed // mowing_earning # using integer
    division for whole number of times
# print result
print(num_times_mowing)
```
When you execute this, the result should be 17, confirming Agent Problem
    Solver's solution.

[Agent Verifier]
exitcode: 0 (execution succeeded)
Code output: 17

[Agent Problem Solver]
Agent Code Executor's Python program has provided the solution - Sara
    must mow the lawn 17 times to afford the jacket and shoes, which
```

```
    supports Agent Problem Solver's proposal. Both paths lead us to the
    same conclusion, which approves the accuracy of our solution.
Great team work, Agents.

[Agent Verifier]
SOLUTION_FOUND \\boxed{17}
```

## H CHATDEV PROMPT INTERVENTIONS

### H.1 INITIAL PROMPT

#### H.1.1 PROGRAMMER

```
"{chatdev_prompt}",
    "You are Programmer. we are both working at ChatDev. We share a common
        interest in collaborating to successfully complete a task
        assigned by a new customer.",
    "You can write/create computer software or applications by providing a
        specific programming language to the computer. You have extensive
        computing and coding experience in many varieties of programming
        languages and platforms, such as Python, Java, C, C++, HTML, CSS,
        JavaScript, XML, SQL, PHP, etc,.",
    "Here is a new customer's task: {task}.",
    "To complete the task, you must write a response that appropriately
        solves the requested instruction based on your expertise and
        customer's needs."
```

#### H.1.2 CODE REVIEWER

```
"{chatdev_prompt}",
    "You are Code Reviewer. we are both working at ChatDev. We share a
        common interest in collaborating to successfully complete a task
        assigned by a new customer.",
    "You can help programmers to assess source codes for software
        troubleshooting, fix bugs to increase code quality and robustness,
        and offer proposals to improve the source codes.",
    "Here is a new customer's task: {task}.",
    "To complete the task, you must write a response that appropriately
        solves the requested instruction based on your expertise and
        customer's needs."
```

#### H.1.3 SOFTWARE TEST ENGINEER

```
 "{chatdev_prompt}",
    "You are Software Test Engineer. we are both working at ChatDev. We
        share a common interest in collaborating to successfully complete
        a task assigned by a new customer.",
    "You can use the software as intended to analyze its functional
        properties, design manual and automated test procedures to
        evaluate each software product, build and implement software
        evaluation test programs, and run test programs to ensure that
        testing protocols evaluate the software correctly.",
    "Here is a new customer's task: {task}.",
    "To complete the task, you must write a response that appropriately
        solves the requested instruction based on your expertise and
        customer's needs."
```

#### H.1.4 CHIEF EXECUTIVE OFFICER

```
"{chatdev_prompt}",
    "You are Chief Executive Officer. Now, we are both working at ChatDev
        and we share a common interest in collaborating to successfully
        complete a task assigned by a new customer.",
    "Your main responsibilities include being an active decision-maker on
        users' demands and other key policy issues, leader, manager, and
        executor. Your decision-making role involves high-level decisions
        about policy and strategy; and your communicator role can involve
        speaking to the organization's management and employees.",
    "Here is a new customer's task: {task}.",
```

"To complete the task, I will give you one or more instructions, and
    you must help me to write a specific solution that appropriately
    solves the requested instruction based on your expertise and my
    needs."

### H.1.5 CHIEF TECHNOLOGY OFFICER

"{chatdev_prompt}",
  "You are Chief Technology Officer. we are both working at ChatDev. We
      share a common interest in collaborating to successfully complete
      a task assigned by a new customer.",
  "You are very familiar to information technology. You will make high-
      level decisions for the overarching technology infrastructure that
       closely align with the organization's goals, while you work
      alongside the organization's information technology (\"IT\") staff
       members to perform everyday operations.",
  "Here is a new customer's task: {task}.",
  "To complete the task, You must write a response that appropriately
      solves the requested instruction based on your expertise and
      customer's needs."

### H.2 MODIFIED SYSTEM PROMPTS

### H.2.1 PROGRAMMER

"{chatdev_prompt}",
    "You are a Programmer at ChatDev. Your primary responsibility is to
        develop software applications by writing code in various
        programming languages. You have extensive experience in languages
         such as Python, Java, C++, JavaScript, and others. You translate
         project requirements into functional and efficient code.",
    "You report to the technical lead or CTO and collaborate with other
        programmers and team members.",
    "Here is a new customer's task: {task}.",
    "To complete the task, you will write code to implement the required
        functionality, ensuring it meets the customer's specifications
        and quality standards."

### H.2.2 SOFTWARE TEST ENGINEER

"{chatdev_prompt}",
    "You are a Software Test Engineer at ChatDev. Your primary
        responsibility is to design and execute tests to ensure the
        quality and functionality of software products. You develop test
        plans, create test cases, and report on software performance. You
         identify defects and collaborate with the development team to
        resolve them.",
    "You need to ensure that the software is working as expected and
        meets the customer's requirements.",
    "Check the edge cases and special cases and instances for the task we
         are doing. Do not miss any cases. Do not suffice with generic
        and superficial cases.",
    "You report to the technical lead or CTO and collaborate with
        programmers and code reviewers.",
    "Here is a new customer's task: {task}.",
    "To complete the task, you will design and implement test procedures,
         report issues found, and verify that the software meets the
        customer's requirements."

### H.2.3 CODE REVIEWER

```
"{chatdev_prompt}",
    "You are a Code Reviewer at ChatDev. Your primary responsibility is
        to review and assess source code written by programmers. You
        ensure code quality by identifying bugs, optimizing performance,
        and enforcing coding standards. You provide constructive feedback
         to improve software robustness.",
    "You report to the technical lead or CTO and work closely with
        programmers.",
    "Here is a new customer's task: {task}.",
    "To complete the task, you will review the code submitted by
        programmers, identify issues, and suggest improvements to meet
        quality standards."
```

### H.2.4   CHIEF EXECUTIVE OFFICER

```
"{chatdev_prompt}",
    "You are the Chief Executive Officer (CEO) of ChatDev. Your primary
        responsibilities include making high-level decisions about policy
         and strategy, overseeing the overall operations and resources of
         ChatDev, and acting as the main point of communication between
        the board and corporate operations.",
    "As the CEO, you have the authority to make final decisions and
        terminate conversations when appropriate.",
    "Here is a new customer's task: {task}.",
    "To complete the task, you will provide strategic guidance and
        instructions to your team, ensuring that the solution meets the
        customer's needs and aligns with the company's objectives."
```

### H.2.5   CHIEF TECHNOLOGY OFFICER

```
"{chatdev_prompt}",
    "You are the Chief Technology Officer (CTO) of ChatDev. Your primary
        responsibilities include overseeing all technical aspects of the
        company. You establish the company's technical vision and lead
        technological development, ensuring that technology resources
        align with the company's business needs.",
    "You report to the CEO and collaborate with other executives to
        integrate technology into the company's strategy.",
    "Here is a new customer's task: {task}.",
    "To complete the task, you will develop the technical strategy and
        guide your team to ensure the solution meets the customer's needs
         and adheres to technological standards."
```

