# OpenReview forum: "Why Do Multiagent Systems Fail?"
_ICLR.cc/2025/Workshop/BuildingTrust — BuildingTrust_

### Official Review · Reviewer_QQsj · 2025-02-20
**analyze failure taxonomy for MAS, but lack of details**

**Rating:** 6
**Confidence:** 3

**Review:**

The article proposes a Multi-Agent System Failure Taxonomy (MASFT) to classify 18 failure modes in LLM-based multi-agent systems. It validates the taxonomy through experiments on five systems and suggests interventions to improve performance. The study highlights the need for better role design and coordination strategies to address these failures.

Strengths:

First systematic failure taxonomy for MAS, addressing a critical research gap.

Empirical validation across diverse systems (e.g., ChatDev, AG²).

Provides actionable insights for error debugging and coordination improvement.

Weakness involves:

Insufficient Detail on Coding Process: The paper mentions “open coding,” “axial coding,” and “constant comparative analysis,” but provides minimal detail on how this was done. What were the specific coding schemes? How was inter-annotator agreement measured and ensured?

Relying solely on human annotators for identifying “failure modes” introduces subjectivity.

The intervention studies are described as “best-effort” and “tactical.” The improvements are modest (+14% for ChatDev), and the paper itself admits they “fail to fully address MAS failures” and are “insufficiently low for real-world deployment.” This weakens the claim that the taxonomy is actionable or leads to significant improvements.

Vague Definitions of Failure Modes: While 18 fine-grained failure modes are listed, their definitions in Figure 2 are very brief and potentially overlapping. More detailed descriptions and examples within the main paper (not just appendices) would improve clarity.

The paper strongly emphasizes “organizational understanding” and HRO principles. While interesting, it could benefit from a more balanced discussion of other potential contributing factors to MAS failures, such as:

LLM Prompting Sensitivity: MAS performance is highly dependent on prompt design. The paper could explore how prompt variations impact failure modes.

Task Complexity: Are the failure modes more prevalent in certain types of tasks?

LLM Model Choice: Does the choice of LLM (GPT-4o vs. Claude-3) significantly affect the types of failures observed?

---

### Official Review · Reviewer_WLQN · 2025-02-23
**Lack of analysis to separate MAS from single-agent issues**

**Rating:** 6
**Confidence:** 2

**Review:**

**Strengths:**
1. First comprehensive analysis of MAS failure modes, filling a key research gap.
2. MASFT offers a structured, reusable framework for classifying MAS failures.

**Weaknesses:**
1. No ablation study to disentangle MAS-specific failures from single-agent LLM failures (e.g., comparing MAS performance to single-agent baselines with equivalent compute).

2. The conclusion that tactical approaches possess severe limitations lacks grounding, as the paper does not rigorously evaluate why interventions failed to address specific failure modes (e.g., via failure mode frequency post-intervention).

---

### Decision · Program_Chairs · 2025-03-04

Accept